# Machine Learning Approaches for Outdoor Air Quality Modelling: A Systematic Review

**Yves Rybarczyk [1,2]** and **Rasa Zalakeviciute [1,*]**

1   Intelligent & Interactive Systems Lab (SI$_2$ Lab), Universidad de Las Américas, 170125 Quito, Ecuador; y.rybarczyk@fct.unl.pt

2   Department of Electrical Engineering, CTS/UNINOVA, Nova University of Lisbon, 2829-516 Monte de Caparica, Portugal

*   Correspondence: rasa.zalakeviciute@udla.edu.ec; Tel.: +351-593-23-981-000

**Abstract:** Current studies show that traditional deterministic models tend to struggle to capture the non-linear relationship between the concentration of air pollutants and their sources of emission and dispersion. To tackle such a limitation, the most promising approach is to use statistical models based on machine learning techniques. Nevertheless, it is puzzling why a certain algorithm is chosen over another for a given task. This systematic review intends to clarify this question by providing the reader with a comprehensive description of the principles underlying these algorithms and how they are applied to enhance prediction accuracy. A rigorous search that conforms to the PRISMA guideline is performed and results in the selection of the 46 most relevant journal papers in the area. Through a factorial analysis method these studies are synthetized and linked to each other. The main findings of this literature review show that: (i) machine learning is mainly applied in Eurasian and North American continents and (ii) estimation problems tend to implement Ensemble Learning and Regressions, whereas forecasting make use of Neural Networks and Support Vector Machines. The next challenges of this approach are to improve the prediction of pollution peaks and contaminants recently put in the spotlights (e.g., nanoparticles).

**Keywords:** atmospheric pollution; predictive models; data mining; multiple correspondence analysis

## 1. Introduction

Worsening air quality is one of the major global causes of premature mortality and is the main environmental risk claiming seven million deaths every year [1]. Nearly all urban areas do not comply with air quality guidelines of the World Health Organization (WHO) [2,3]. The risk populations that suffer from the negative effects of air pollution the most are children, elderly, and people with respiratory and cardiovascular problems. These health complications can be avoided or diminished through raising the awareness of air quality conditions in urban areas, which could allow citizens to limit their daily activities in the cases of elevated pollution episodes, by using models to forecast or estimate air quality in regions lacking monitoring data.

Air pollution modelling is based on a comprehensive understanding of interactions between emissions, deposition, atmospheric concentrations and characteristics, meteorology, among others; and is an indispensable tool in regulatory, research, and forensic applications [4]. These models calculate and predict physical processes and the transport within the atmosphere [5]. Therefore, they are widely used in estimating and forecasting the levels of atmospheric pollution and assessing its impact on human and environmental health and economy [6–9]. In addition, air pollution modelling is used in science to help understand the relevant processes between emissions and concentrations, and understand the interaction of air pollutants with each other and with weather [10]

and terrain [11,12] conditions. Modelling is not only important in helping to detect the causes of air pollution but also the consequences of past and future mitigation scenarios and the determination of their effectiveness [4].

There are a few main approaches to air pollution modelling—atmospheric chemistry, dispersion (chemically inert species), and machine learning. Different complexity Gaussian models (e.g., AERMOD, PLUME) are widely used by authorities, industries, and environmental protection organizations for impact studies and health risk investigations for emissions dispersion from a single or multiple point sources (also line and area sources, in some applications) [13,14]. These models are based on assumptions of continuous emission, steady-state conditions and conservation of mass. Lagrangian models study a trajectory of an air parcel, the position and properties of which are calculated according to the mean wind data over time (e.g., NAME) [5,14]. On the other hand, Eulerian models use a gridded system that monitors atmospheric properties (e.g., concentration of chemical tracers, temperature and pressure) in specific points of interest over time (e.g., Unified Model). Chemical Transport Models (CTMs) (e.g., air-quality, air-pollution, emission-based, source-based, source, etc.) are prognostic models that process emission, transport, mixing, and chemical transformation of trace gases and aerosols simultaneously with meteorology [15]. Complex and computationally costly CTMs can be of a global (e.g., online: Fim-Chem, AM3, MPAS, CAM-Chem, GEM-MACH, etc.; and offline: GEOS-Chem, MOZART, TM5) and regional (e.g., online: MM5-Chem, WRF-Chem, BRAMS; and offline: CMAQ, CAMx, CHIMERE) scale [16,17].

These models combine atmospheric science and multi-processor computing techniques, highly relying on considerable resources like real-time meteorological data and an updated detailed emission sources inventory [18]. Unfortunately, emission inventory inputs for boundary layers and initial conditions may be lacking in some regions, while geophysical characteristics (terrain and land use) might further complicate the implementation of these models [19]. To deal with complex structure of air flows and turbulence in urban areas Computer Fluid Dynamics (CFD) methods are used [20]. However, recent studies show that the traditional deterministic models struggle to capture the non-linear relationship between the concentration of contaminants and their sources of emission and dispersion [21–24], especially in a model application in regions of complex terrain [25]. To tackle the limitations of traditional models, the most promising approach is to use statistical models based on machine learning (ML) algorithms.

Statistical techniques do not consider physical and chemical processes and use historical data to predict air quality. Models are trained on existing measurements and are used to estimate or forecast concentrations of air pollutants according to predictive features (e.g., meteorology, land use, time, planetary boundary layer, elevation, human activity, pollutant covariates, etc.). The simplest statistical approaches include Regression [26], Time Series [27] and Autoregressive Integrated Moving Average (ARIMA) [28] models. These analyses describe the relationship between variables based on possibility and statistical average. Well-specified regressions can provide reasonable results. However, the reactions between air pollutants and influential factors are highly non-linear, leading to a very complex system of air pollutant formation mechanisms. Therefore, more advanced statistical learning (or machine learning) algorithms are usually necessary to account for a proper non-linear modelling of air contamination. For instance, Support Vector Machines [29], Artificial Neural Networks [30], and Ensemble Learning [31] have been applied to overcome non-linear limitations and uncertainties to achieve better prediction accuracy. Although statistical models do not explicitly simulate the environmental processes, they generally exhibit a higher predictive performance than CTMs on fin spatiotemporal scales in the presence of extensive monitoring data [32–34].

Different machine learning approaches have been used in recent years to predict a set of air pollutants using different combinations of predictor parameters. However, with a growing number of studies, it is puzzling why a certain algorithm is chosen over another for a given task. Therefore, in this study we aim to review recent machine learning studies used in atmospheric pollution research. To do so, the remainder of the paper is organized into three sections. First, we explain the method used to select

and scan relevant journal articles on the topic of machine learning and air quality, which conforms to the PRISMA guideline. This section also describes the strategy used to analyze the manuscripts and synthetize the main findings. Second, the results are presented and discussed from a general to a detailed and synthetic account. Finally, the last section draws conclusions on the use of machine learning algorithms for predicting air quality and the future challenges of this promising approach.

## 2. Method

### 2.1. Search Strategy

Relevant papers were researched in SCOPUS. The enquiry was limited to this scientific search engine, because it compiles, in a single database, manuscripts which are also indexed in the most significant databases in engineering (e.g., IEEE Xplore, ACM, etc.). The first step of the literature review consisted of completing the *website document search* with a combination of keywords. The used formula was as follows: {'Machine Learning'} AND {'Air Quality' OR 'Air Pollution'} AND {'Model OR 'Modelling'}. The exploration was limited to the period of 2010–2018. Another limitation was to focus only on journal papers, since they represent the most achieved work. The result of this first step provides us with 103 documents.

The second step consisted of filtering the studies, by reading the title and the abstract. Papers were excluded from our selection if they addressed the topics as follows: Physical sensors (and not computational models); health/epidemiological studies (i.e., predictive models to estimate the impact on health and not to estimate and/or forecast the concentration of pollutants); social studies; biological studies; indoor studies; sporadic calamity (e.g., smog disaster). After applying these rejection criteria, the documents were reduced to 50.

In the last step, all 50 papers were fully read. After a consensus between the authors of this systematic review, four papers were rejected. Three manuscripts were excluded, because they represented a very similar study that was previously carried out by the same authors. The other paper was consensually considered out of scope after reviewing the full document. Consequently, a total of 46 manuscripts were included for a further qualitative and quantitative synthesis. Figure 1 represents the flow diagram of the search method for the systematic review. It is based on the PRISMA approach that provides a guideline to identify, select, assess and summarize the findings of similar but separate studies. This method promotes a quantitative synthesis of the papers, which is carried out through a factorial analysis.

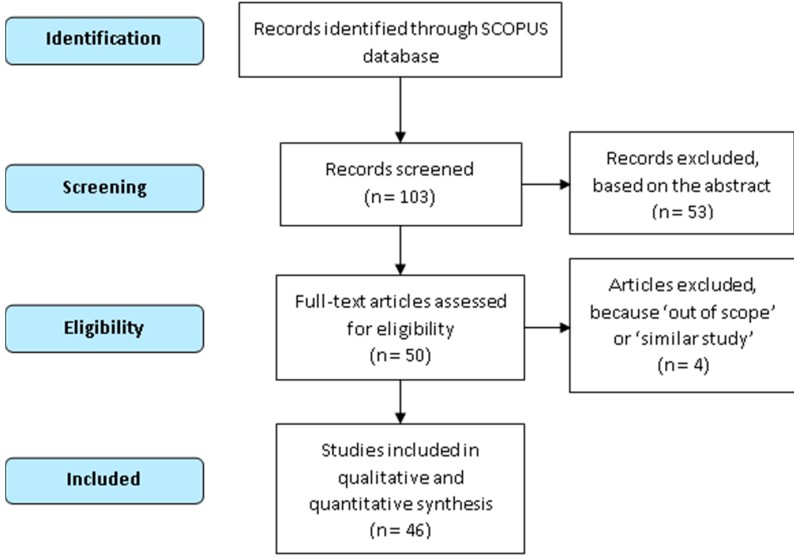

**Figure 1.** PRISMA-based flowchart of the systematic selection of the relevant studies.

*2.2. Analyzed Parameters*

All the papers are analyzed according to 14 aspects. The first parameter concerns the motivation of the study. The second is the type of modelling, which is divided into two categories—estimation and forecasting models. An estimation model uses predictive features (e.g., contaminant covariates, meteorology, etc.) to estimate the concentration of a determined pollutant at the same time. A forecasting model takes into account historical data to predict the concentration of a pollutant in the future. The third analysis is based on the type of machine learning algorithm used. The main categories are artificial neural networks, support vector machines, ensemble learning, regressions, and hybrid versions of these algorithms. The fourth analysis describes the method applied by the authors. The fifth point focuses on the nature of the predicted parameter. Again, two groups are identified. On the one hand, there are authors interested in predicting specific air contaminants, which are: Micro and nano-size particulate matter ($PM_{10}$, $PM_{2.5}$, $PM_1$); nitrogen oxides ($NO_x = NO + NO_2$); Sulphur oxides ($SO_x$); carbon monoxide ($CO$); and ozone ($O_3$). On the other hand, some authors work on a prediction of air quality in general by searching for the Air Quality Index (AQI), which may include the concentrations of several pollutants. The sixth parameter identifies the geographic location of the study. The seventh gives details of the characteristics of the dataset, such as—time span; quantity of monitoring stations; and number of instances. Furthermore, the eighth point provides information on the specificity of the dataset in terms of the used predictive attributes. The main features are related to—pollutant covariates; meteorology; land use; time; human activity; and atmospheric phenomena. The ninth and tenth factors address the evaluation method and the performance of the tested algorithms, respectively. The assessment is mainly based on a comparison of the accuracy between the models and/or a comparison of the prediction of the actual value. The most popular evaluation criteria are the ratio of correctly classified instances (Accuracy), the Mean Absolute Error (MAE), the Root Mean Square Error (RMSE), and the coefficient of determination ($R^2$). The Accuracy represents the overall performance of a classifier by providing the proportion of the whole test set that is correctly classified, as described in Equation (1).

$$\text{Accuracy} = \frac{\text{TP} + \text{TN}}{\text{TP} + \text{TN} + \text{FP} + \text{FN}} \tag{1}$$

where TP, TN, FP and FN stand for True Positives, True Negatives, False Positives and False Negatives, respectively. The higher the Accuracy value is, the better is the model performance. The MAE shows the degree of difference between the predicted values and the actual values. The RMSE is another relative error estimator that focuses on the impact of extreme values based on the MAE. The $R^2$ represents the fitting degree of a regression. The MAE, RMSE and $R^2$ are calculated according to Equations (2)–(4), respectively.

$$\text{MAE} = \frac{1}{n} \sum_{i=1}^{n} |E_i - A_i| \tag{2}$$

$$\text{RMSE} = \left( \frac{1}{n} \sum_{i=1}^{n} (E_i - A_i)^2 \right)^{\frac{1}{2}} \tag{3}$$

$$R^2 = \left( \frac{\sum_{i=1}^{n} (A_i - \overline{A}) \, (E_i - \overline{E})}{\sqrt{\sum_{i=1}^{n} (A_i - \overline{A})^2 \, \sum_{i=1}^{n} (E_i - \overline{E})^2}} \right)^2 \tag{4}$$

where n is the number of instances, $A_i$ and $E_i$ are the actual and estimated values, respectively. $\overline{A}$ and $\overline{E}$ stand for the mean measured and mean estimated value of the contaminant, respectively. $A_{max}$ and $A_{min}$ are the maximum and minimum observed pollutant values, respectively. $R^2$ is a dimensionless descriptive statistical index ranging from 0 (no correlation) to 1 (perfect correlation). MAE and RMSE

values are in # $cm^{-3}$. The lower the MAE and RMSE is, the better is the predictive performance of the model.

The eleventh parameter considers the computational cost of the method. Finally, the last three points discuss the outcomes of the proposed approach, its limitation, and its scope of applicability.

*2.3. Synthesis of Results*

The results are synthetized according to descriptive statistics and a factorial analysis. First, we describe the most used algorithms over time in order to define the current tendencies. Second, we quantify the types of algorithms applied for the prediction of the principal contaminants. Third, we identify the evolution of the modelling performance for each pollutant over the last decade. Finally, since several parameters are considered for the description of the selected papers, we perform a factorial analysis to summarize the main outcomes of this systematic review. The most appropriate method to identify the relationships between the qualitative factors that characterize each study is the Multiple Correspondence Analysis (MCA). First, a data table is created from the parameters identified in Section 2.2 (see Figure A1 of Appendix A). Each row (i) corresponds to each paper and each column (j) corresponds to each qualitative variable (e.g., types of modelling, algorithms, etc.). The total number of papers and qualitative variables are identified as I and J, respectively. Next, this data table is fed to the R software, in order to proceed with the analysis through the MCA library. The initial table is transformed into an indicator matrix called the Complete Disjunctive Table, in which rows are still the papers, but the columns are now the categories (e.g., estimation model vs. forecasting model) of the qualitative variables. The total number of categories is identified as $k_j$. The entry of the intersection of the i-th row and the k-th column, which is called $y_{ik}$, is equal to 1 (or true) if the i-th paper has category k of the j-th variable and 0 (or false) otherwise. All the papers have the same weight, which is equal to $1/I$. However, the method highlights papers that are characterized by rare categories by implementing Equation (5).

$$x_{ik} = \frac{y_{ik}}{p_k} \tag{5}$$

where $p_k$ represents the proportion of papers in the category k. Then, the data are centralized by applying Equation (6).

$$x_{ik} = \frac{y_{ik}}{p_k} - 1 \tag{6}$$

The table of the $x_{ik}$ is used to build the points cloud of papers and categories. Since, the variables are centered, the cloud has a center of gravity in the origin of the axes. The distance between a paper i and a paper i' is given by Equation (7).

$$d_{i,i'}^2 = \sum_{k=1}^{K} \frac{p_k}{J} \left( x_{ik} - x_{i'k} \right)^2 = \frac{1}{J} \sum_{k=1}^{K} \frac{1}{p_k} \left( y_{ik} - y_{i'k} \right)^2 \tag{7}$$

where $x_{ik}$ and $x_{i'k}$ are the coordinates of the papers i and i', respectively. This equation shows that the distance is 0 if two papers are in the same set of categories (or profile). If two papers share many categories, the distance will be small. However, if two papers share several categories except a rare one, the distance between them will be relatively large due to the $p_k$ value. Next, the calculation of the distance of a point from the origin (O) is given by Equation (8).

$$d(i, O)^2 = \sum_{k=1}^{K} \frac{p_k}{J} (x_{ik})^2 = \frac{1}{J} \sum_{k=1}^{K} \frac{y_{ij}}{p_k} - 1 \tag{8}$$

The equation shows that a distance gets larger when the categories of a paper are rarer, because its $p_k$ are small. In other words, the more a paper possesses rare categories, the further it is from the origin of the plot axes. To conclude the process, the point cloud must be projected into a smaller-dimensional

space, which is usually reduced to two dimensions. To do so, the cloud ($N_I$) is projected onto a sequence of orthogonal axes with maximum inertia as calculated in Equation (9).

$$\text{Inertia}(N_I) = \max \sum_{i=1}^{I} \frac{1}{I} d^2(i, O) \tag{9}$$

In terms of the first two dimensions obtained, we end up with a factor plane for the papers and categories that is the best bi-dimensional representation of them.

## 3. Results and Discussion

This section is organized in three parts. First, a general description of the results of the systematic review is provided. These results are mainly based on descriptive statistics regarding the number of studies that uses a machine learning approach over the last decade, the geographic distribution of these studies across the world, and most popular algorithms implemented. The second part provides a detailed development of the 46 papers selected following the method defined in Section 2.1. These papers are grouped into six categories for a more comprehensive description. The definition of the groups is based on the principal motivation of the study. Category 1 groups the manuscripts that are interested in identifying the most relevant predictors and understanding their non-linear relationship with air pollution. Category 2 corresponds to a set of papers that deals with image-based monitoring and the way to tackle low spatial resolution from non-specific/low resolution sensors. Category 3 is specialized on a family of studies that considers land use and/or spatial heterogeneity as a feature. Category 4 focuses on hybrid models as well as extreme and deep learning. Category 5 is a bunch of work, which resulted in an applicative system. Finally, Category 6 addresses the new challenge of predicting the concentration of nanoparticle matter.

### 3.1. General Description

Figure 2 shows the evolution of the number of journal papers using a machine learning approach to model air quality, since 2010. The number of studies in this field shows an increasing trend over the last eight years. An important boom can be observed from 2015. The average number of studies has been multiplied by a factor of 5, since the period 2011–2015. This high value tends to remain stable during the last two years. It is of note that the literature review was performed in September 2018, which supposes a possibly superior number of documents by the end of the present year.

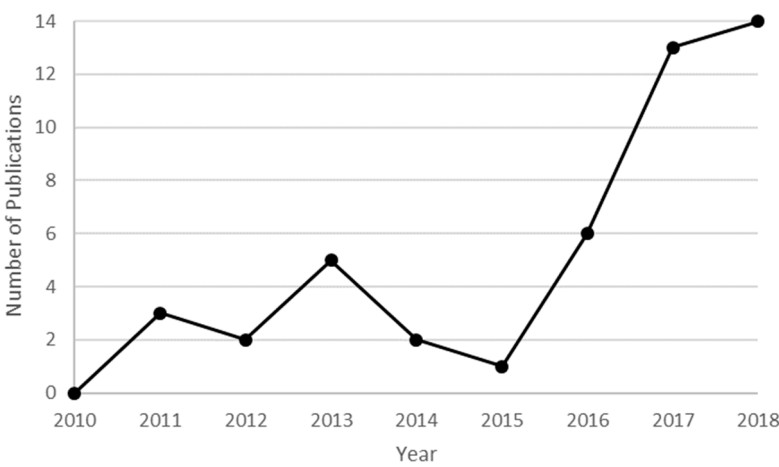

**Figure 2.** The evolution of published studies in scientific journals regarding machine learning techniques used in atmospheric sciences.

While the number of machine learning approaches for predicting atmospheric pollutants has increased dramatically, this growth has not been uniformed globally. Figure 3 shows that the number of studies is much higher in the northern hemisphere, specifically in Eurasia and North America. In addition, the more complex list of pollutants has been studied in Asia and Europe, including all criteria pollutants and even a specific pollutant or an overall Air Quality Index (AQI). While the biggest number of studies has been concentrated in China (13 studies) and United States (seven studies), other parts of the world have more limited published research. Especially unstudied regions of the world are South America and Oceania, both with one study, focusing on a single specific pollutant.

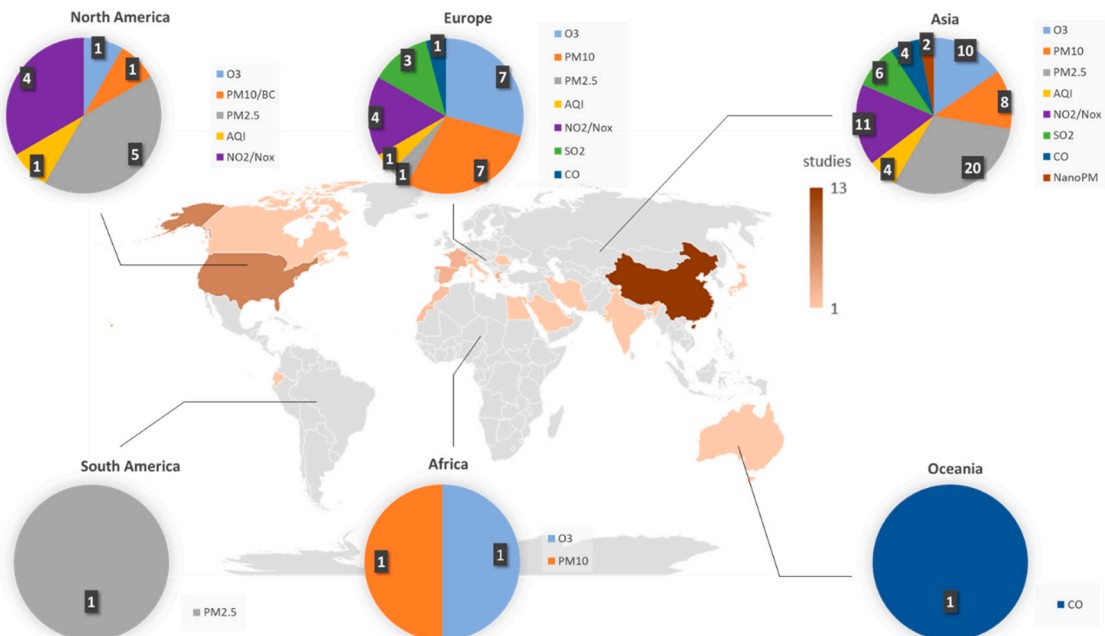

**Figure 3.** Number of papers per country and types of atmospheric contaminants studied by continent. The world map is colored by number of studies per country (yellow–brown), while the grey color indicates regions with no studies. The pollutant studies were considered per continent: $O_3$—ozone, $PM_{10}$—particulate matter with aerodynamic diameters $\leq 10$ μm, BC—black carbon, $PM_{2.5}$—particulate matter with aerodynamic diameters $\leq 2.5$ μm, AQI—Air Quality Index, $NO_2$—nitrogen dioxide, $NO_x$—nitrogen oxides ($NO + NO_2$), $SO_2$—sulfur dioxide, CO—carbon dioxide, NanoPM—nano particles (particles with diameter in nm).

The systematic review shows that the most used algorithms in descending order are: Ensemble Learning Methods, Artificial Neural Networks, Support Vector Machines, and Linear Regressions (Figure 4). Two papers use more peculiar approaches related to Decision rules and Lazy methods. Ensemble Learning is a class of machine learning techniques that creates multiple predictors to address the same problem and make a single prediction by combining the results from the predictors. Thus, the result of the predictive model is obtained by taking an average or the majority voting. The multiple predictors are created by introducing stochasticity into the data or the prediction algorithm. In the first case, different datasets and respective predictors are produced from the original database. In the second case, the randomness is introduced by using different types of algorithms (e.g., Decision Trees and Neural Network) to solve a single problem. The advantage of using an Ensemble Learning method is to provide a better accuracy, if we compare to the performance of each predictor taken individually (also called weak predictors). Bagging (or Bootstrap Aggregation) is the simplest way to introduce stochasticity in the data. In this technique, several datasets are created by taking random samples with replacement. Then, one specific predictor is produced for each dataset, and theses predictors are aggregated to give the final model. Boosting is another way to create stochasticity by sequentially training each weak classifier. These predictors are weighted in

some way that is usually related to the accuracy of each predictor. It means that observations which are incorrectly predicted get a higher weight. So, the best predictor is the one that has to perform better on these critical observations. Finally, Stacking is the method used to make a prediction from a battery of different possible algorithms. After applying these algorithms to the dataset, they are assessed by cross-validation, for instance, in order to give a weight to each method according to its performance. Random Forest is one of the most popular Ensemble Learning techniques and constitutes the majority of the papers classified under this label in our literature review. This method is based on an ensemble of Decision Tree predictors. The first step implements bagging on trees. Then, a second step adds stochasticity to split the tree by using the random subspace method or the random split selection, which consists of applying at each node the algorithm with a subset only of the features to split the node.

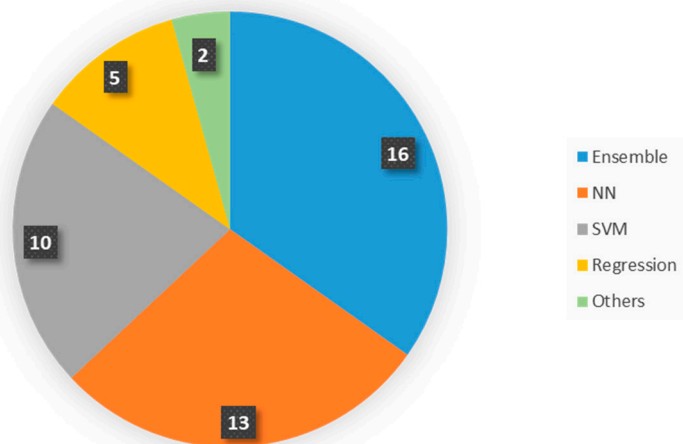

**Figure 4.** Proportion of machine learning algorithms used. 'Ensemble' stands for Ensemble Learning (mainly Random Forest). 'NN' stands for any algorithm based on an Artificial Neural Networks approach. 'SVM' stands for Support Vector Machine (it also includes Support Vector Regression). 'Regression' stands for Multiple Linear Regression. 'Others' includes Decision Rules and Lazy methods.

Artificial Neural Networks (NN) are inspired by the working of the human being's nervous system. The simplest neural network architecture is called Perceptron. This method classifies inputs by using a linear combination of the observations of the dataset, as defined in Equation (10).

$$x = \sum_{j=0}^{J} w_j o_j \tag{10}$$

where o is an observation characterized by several features (from 0 to J) and $w_j$ is the weight applied to each feature. For instance, in the case of two classes, if x is greater than 0 the observation is classified in class 1, otherwise it is classified in class 2. In an NN, the weights are learned. The algorithm starts by setting all weights to 0. Then, a loop is applied until all observations in the training data are classified correctly. In this loop, for each observation o, if o is classified incorrectly and if it belongs to class 1, it is added to the weight vector, otherwise it is subtracted. No modification is applied if the observation is classified correctly. Nevertheless, the NN implemented in the selected papers consist of at least three types of layers (i.e., Multilayer Perceptron)—input layer, hidden layer, and output layer. The input is composed with the features (e.g., contaminant covariates, meteorology, etc.). The output is the variable to be predicted (i.e., concentration of a contaminant or value of an air quality index). And the hidden layer consists of additional nodes between the two previous layers that enables multiple kinds of connections. When the number of hidden layers is large the NN is called Deep Learning Neural Network. Each node (or neuron) performs a weighted sum of its input and thresholds

the result. Instead of using the binary threshold of the basic Perceptron, the Multilayer Perceptron usually applies a smooth activation function based on a sigmoid. In such an implementation, as the inputs become more extreme, they approach the step function, which corresponds to the hard-edge threshold used in the basic Perceptron. Finally, the learning stage consists of getting the best weights to minimize the error between the predictive output and the actual output. This is an iterative process using a steepest descent method, in which the gradient is determined by a backpropagation algorithm that seeks to minimize the cost function $J(\theta)$ defined in Equation (11).

$$J(\theta) = \frac{1}{m} \sum_{i=1}^{m} \left( h_\theta \left( x^{(i)} \right) - y^{(i)} \right)^2 \tag{11}$$

where m is the number of observations, y is the actual output and $h_\theta(x)$ is the predicted output.

Support Vector Machine (SVM) is another algorithm that draws a boundary through the widest channel between two classes, which is the maximum separation from each class. This boundary line is obtained by selecting the critical points (or support vectors) that define the channel and taking the perpendicular bisector of the line joining those two support vectors. In the case of a multidimensional dataset the boundary is defined by a hyperplane. Equation (12) shows that the calculation of the maximum margin hyperplane depends only on the value of the support vectors and not the remaining observations.

$$x = b + \sum_{i=0}^{I} \alpha_i y_i a(i) \times a \tag{12}$$

where a(i) are the support vectors and I is the number of these vectors. In cases where classes are not linearly separable, a function is used to transform the data into a high dimensional space (e.g., a polynomial function). A kernel trick is used to reduce the computational cost of this transformation. The most current kernel types are —linear kernel (Equation (13)), polynomial kernel (Equation (14)), and radial basis function kernel (Equation (15)).

$$K(x, y) = x \times y \tag{13}$$

$$K(x, y) = (x \times y + 1)^d \tag{14}$$

$$K(x, y) = e^{-\gamma ||x-y||^2} \tag{15}$$

It is important to choose the right kernel and to tune its parameters correctly to get a good performance from a classifier. A usual parameter tuning technique includes k-fold cross-validation. SVM is a popular alternative to artificial neural networks.

Finally, a fair number of papers applies a Regression method as machine learning algorithm to predict air quality. Features derived from the dataset are used as input of the Regression model to predict continuous valued output. As in the NN approach, the prediction is obtained by learning the relationship (or weights) between the inputs and the output. These weights are acquired by fitting a linear or nonlinear curve to the data points. In order to correctly fit the curve, it is necessary to define the goodness-of-fit metric, which allows us to identify the curve that fits better than the other ones. The optimization technique used in regression, and in several other machine learning methods (e.g., NN), is the gradient descent algorithm that aims to minimize the cost function $J(\theta)$ defined in Equation (11).

### 3.2. Detailed Description

This section consists of a detailed explanation of the selected papers. It is organized into six categories based on the main motivation of the study. In each group, we first present manuscripts on estimation models and, second, studies that involve forecasting models. An estimation model is defined as an approach that seeks to estimate the value of a pollutant (or index of contamination) from

the measurement of other predictive parameters at the same time step. It is mostly used to give an approximation of the concentrations on an extended geographic area. On the contrary, the objective of a forecasting model is to predict the concentration levels of the contaminants in the near future. We call both estimation and forecasting prediction or predictive models.

3.2.1. Category 1: Identifying Relevant Predictors and Understanding the Non-Linear Relationship with Air Pollution

Category 1 is the biggest group and accounts for total of 16 articles—twelve in estimation modeling and four in forecasting. In the estimation modeling there are five cases of Random Forest (RF) or tree-based Ensemble learning algorithms (e.g., M5P), four regression models, one lazy learning, and two SVM. The latter was once used in forecasting studies, NN was used twice and M5P once. This category highly concentrates on identifying the most contributing parameters to a successful prediction of atmospheric pollution. A variety of tests are used to differentiate between the influential predictors, like Principal Component Analysis (PCA), Quantile Regression Model (QRM), Pearson correlation, etc.

Among the most recent studies we find Grange et al. [22]. This study aims to (i) build a predictive model of $PM_{10}$ based on meteorological, atmospheric, and temporal factors and (ii) analyze $PM_{10}$ trend during the last ten years. The Random Forest model, a simple, efficient, and easy interpretable method, is used. First of all, many out-of-bag samples (randomly sampling observations with replacement and sampling of feature variables) of the training set are used to grow different Decision Trees (DT), and then all the trees are aggregated to form a single prediction. The algorithms are trained and run on 20 years of data in Switzerland from 31 monitoring stations. Daily average data of meteorology (wind speed, wind direction, and temperature), synoptic scale, boundary layer height, and time are used. The model is validated by comparison between the observed and predicted value, resulting in average value in 31 sites: $r^2 = 0.62$. The best predictors are—wind speed, day of the year (seasonal effect), and boundary layer height, whereas the worst predictors are day of the week, and synoptic scale. The model performance is low and the predictive accuracy is inconsistent ($0.53 < r^2 < 0.71$) for different locations, especially varying for the rural mountain sites.

Another study also relies on RF to build a spatiotemporal model to predict $O_3$ concentration across China [35]. First the RF model is built by averaging the predictions from 500 decision/regression trees, and then Random Forest Regressor from the python package scikit-learn is used to run the algorithm. For that meteorology, planetary boundary height, elevation, anthropogenic emission inventory, land use, vegetation index, road density, population density, and time are used from 1601 stations over one year in all of China. The model is validated by comparison of the $r^2$ and RMSE (predicted value/actual value) to Chemical Transport Models, and shows a good performance of $r^2 = 0.69$ (specifically, $r^2 = 0.56$ in winter, and $r^2 = 0.72$ in fall). Interestingly, the model results are quite comparable or even higher than the predictive performance of the CTMs for lower computational costs. While for the predictive features, meteorological factors account for 65% of the predictive accuracy (especially humidity, temperature, and solar radiation), also showing a higher predictive performance when the weather conditions are stronger (i.e., autumn). In this study, anthropogenic emissions ($NH_3$, CO, Organic Carbon, and $NO_x$) exhibit a lower importance than meteorology and lower accuracy is registered for the regions with a sparser density of monitoring stations. Therefore, the accuracy relies on the complexity of the network.

Martínez-España et al. [36] aim to identify the most robust machine learning algorithms to preserve a fair accuracy when $O_3$ monitoring failures may occur, by using five different algorithms—Bagging vs. Random Committee (RC) vs. Random Forest vs. Decision Tree vs. k-Nearest Neighbors (kNN). First, the prediction accuracy of the five machine learning algorithms is compared, then a hierarchical clustering technique is applied to identify how many models are needed to predict the $O_3$ in the region of Murcia, Spain. Two years of pollutant covariates ($O_3$, NO, $NO_2$, $SO_2$, $NO_x$, $PM_{10}$, $C_6H_6$, $C_7H_8$, and XIL) and meteorological parameters (temperature, relative humidity, pressure,

solar radiation, wind speed and direction) are used. The model is validated by the comparison of the coefficient of determination (predicted value/actual value) between the five models. Random Forest slightly outperforms the other models ($r^2 = 0.85$), followed by RC ($r^2 = 0.83$), Bagging and DT ($r^2 = 0.82$) and kNN ($r^2 = 0.78$). The best predictors are $NO_x$, temperature, wind direction, wind speed, relative humidity, $SO_2$, NO, and $PM_{10}$. Finally, the hierarchical cluster shows that two models are enough to describe the studied regions.

Bougoudis et al. [37] aim to identify the conditions under which high pollution emerges and to use a better generalized model. Hybrid system based on the combination of unsupervised clustering, Artificial Neural Networks (ANN) and Random Forest (RF) ensembles and fuzzy logic is used to predict multiple criteria pollutants in Athens, Greece. Twelve years of hourly data of CO, NO, $NO_2$, $SO_2$, temperature, relative humidity, pressure, solar radiation, wind speed and direction are used. Unsupervised clustering of the initial dataset is executed in order to re-sample the data vectors; while ensemble ANN modeling is performed using a combination of machine learning algorithms. The optimization of the modeling performance is done with Mamdani rule-based fuzzy inference system (FIS, used to evaluate the performance of each model) that exploits relations between the parameters affecting air quality. Specifically, self-organizing maps are used to perform dataset re-sampling, then ensembles of feedforward artificial neural networks and random forests are trained to clustered data vectors. The estimation model performance is quite good for CO ($r^2 = 0.95$, RF_FIS), NO ($r^2 = 0.95$, ensemble NN_FIS), NO and $O_3$ ($r^2 = 0.91$, RF) and $SO_2$ ($r^2 = 0.78$, ensemble regression).

Sayegh et al. [38] also employ a number of models (Multiple Linear Regression vs. Quantile Regression Model (QRM) vs. Generalized Additive Model vs. Boosted Regression Trees) to perform a comparative study on the performance for capturing the variability of $PM_{10}$. At the contrary of the linear regression, which considers variables distribution as a whole, the QRM defines the contribution (coefficient) of the predictors for different percentiles of $PM_{10}$ (here 10 percentiles or quantiles are used) to estimate the weight of each feature. Meteorological factors (wind speed, wind direction, temperature, humidity), chemical species (CO, $NO_x$, $SO_2$), and $PM_{10}$ of the previous day data for one year from Makkah (Saudi Arabia) are used. The model performance is validated with observed data and the QRM model shows a better performance ($r^2 = 0.66$) in predicting hourly $PM_{10}$ concentrations due to the contribution of covariates at different quantiles of the dependent variable ($PM_{10}$), instead of considering the central tendency of $PM_{10}$.

Singh et al. [39] aim to identity sources of pollution and predict the air quality by using Hybrid Model of Principal Components Analysis, Tree-based Ensemble Learning (Single Decision Tree (SDT), Decision Tree Forest (DTF) and Decision Tree Boost (DTB)) vs. Support Vector Machine Model. Five years of air quality and meteorological parameter data are used for Lucknow (India). SDT, DTF, DTB and SVM are used to predict the Air Quality Index (AQI), and Combined AQI (CAQI)), and to determine the importance of predictor features. The model performance is validated by a comparison between the models and with the observed data. Decision Tree models—SDT ($r^2 = 0.9$), DTF ($r^2 = 0.95$), and DTB at $r^2 = 0.96$) outperform SVM ($r^2 = 0.89$).

Philibert et al. [40] aim to predict a greenhouse gas $N_2O$ emission using Random Forest vs. Two Regression models (linear and nonlinear) by employing global data of environmental and crop variables (e.g., fertilization, type of crop, experiment duration, country, etc.). The extreme values from data are removed, excluding the boreal ecosystem; input variable features are ranked by importance; controlling number of input variables to result in better prediction. The model is validated by a comparison to the regression model and the simple non-linear model (10-fold cross validation). RF outperforms the regression models by 20–24% of misclassifications.

Meanwhile, Nieto et al. [41] aim to predict air pollution ($NO_2$, $SO_2$ and $PM_{10}$) in Oviedo, Spain based on a number of factors using Multivariate Adaptive Regression Splines and ANN Multilayer Perceptron (MLP). Three years of $NO_x$, CO, $SO_2$, $O_3$, and $PM_{10}$ data modeling result in a good estimation of $NO_2$: $r^2 = 0.85$; $SO_2$: $r^2 = 0.82$; $PM_{10}$ $r^2 = 0.75$ when compared with observed data.

Kleine Deters et al. [42] aim to identify the meteorology effects on $PM_{2.5}$, by predicting it using six years of meteorological data (wind speed and direction, and precipitation) for Quito, Ecuador in regression modeling. This is a good simplified technique and an economic option for the cities with no air quality equipment. The model gets validated by regression between observed and predicted $PM_{2.5}$, and by a 10-fold cross validation of predicted vs. observed $PM_{2.5}$, which shows that this method is a fair approach to estimate $PM_{2.5}$ from meteorological data. In addition, the model performance is improved in the more extreme weather conditions (most influential parameters), as previously seen in Zhan et al. [35].

Carnevale et al. [43] use the lazy learning technique to establish the relationship between precursor emissions ($PM_{10}$) and pollutants (AQI of $PM_{10}$) for the Lombard region (Italy), using hourly Sulfur Dioxide ($SO_2$), Nitrogen oxides ($NO_x$), Carbon Monoxide (CO), $PM_{10}$, $NH_3$ data for one year. The Dijkstra algorithm is deployed in the large-scale data processing system. Eighty percent of the data are used as an example set and 20% as validation set (every 5th cell). The validation phase of surrogate models is performed comparing the output to deterministic model simulations, not the observations. Constant, linear, quadratic and combination approximation based on elevation of the area (0, 1, 2 and all polynomial approximations) are applied. The performance of the model is very comparable ($r^2 = 0.99$–1) to the Transport Chemical Aerosol Model (TCAM), a model that is computationally much costlier, currently used in decision making.

Suárez Sánchez et al. [44] on the other hand, aim to estimate the dependence between primary and secondary pollutants; and most contributing factors in air quality using SVM radial (Gaussian), linear, quadratic, Pearson VII Universal Kernels (PUK) and multilayer perceptron (MLP) to predict $NO_x$, CO, $SO_2$, $O_3$, and $PM_{10}$. Three years of NO, $NO_2$, CO, $SO_2$, $O_3$, and $PM_{10}$ data are used in Aviles, Spain. The model is 10-fold cross validated with the observed data, resulting in best performance using PUK for $NO_x$ and $O_3$ ($r^2 = 0.8$).

Liu et al. [23] also employ SVM to get the most reliable predictive model of air quality (AQI) by considering monitoring data of three cities in China (Beijing, Tianjin, and Shijiazhuang). Two years of last-day AQI values, pollutant concentrations ($PM_{2.5}$, $PM_{10}$, $SO_2$, CO, $NO_2$, and $O_3$), meteorological factors (temperature, wind direction and velocity), and weather description (ex. cloudy/sunny, or rainy/snowy, etc.) are used as predicting features. The dataset is split into a training set and testing set through a 4-fold cross validation technique. To validate the model performance, the results are compared to the observed data. The model performance is especially improved when the surrounding cities' air quality information is included.

Vong et al. [45] also use SVM to forecast air quality ($NO_2$, $SO_2$, $O_3$, SPM) from pollutants and meteorological data in Macau (China). The training is performed on three years of data and tested on one whole year and then just on January and July for seasonal predictions. The Pearson correlation is used to identify the best predictors for each pollutant and different kernels are used to test which of the predictors or models get the best results. The Pearson correlation is also employed to determine how many days of data are optimal for forecasting. The model performance is validated by comparison to the observed data resulting in best kernel—linear and RBF, in polynomial (summer), confirming that SVM performance depends on the appropriate choice of the kernel.

In the forecasting model category, there are two articles that use NN, and meteorology and pollutants as predicting features. Chen et al. [46] aim to build a model that forecasts AQI one day ahead by using an Ensemble Neural Network that processes selected factors using PEK-based machine learning for 16 main cities in China (three years of data). First a selection of the best predictors ($PM_{2.5}$, $PM_{10}$, and $SO_2$) is performed, based on Partial Mutual Information (PMI), which measures the degree of predictability of the output variable knowing the input variable, and then the daily AQI value is predicted, through PEK-based machine learning, by using the previous day's meteorological and pollution conditions. The model is validated by a comparison between actual and predicted value, resulting in an average value of $r^2 = 0.58$.

Meanwhile, Papaleonidas and Iliadis [47] present the spatiotemporal modelling of $O_3$ in Athens (Greece) also using NN, specifically developing 741 ANN (21 years of data). A multi-layer feed forward and back propagation optimization are employed. The best results are conceived with three approaches—the simplest ANN, the best ANN for each case of predictions, and the dynamic variation of NN. The performance of the model is validated with the observed data and between NN approaches, resulting in $r^2 = 0.799$, performing the best for the surrounding stations. This approach is concluded as a good option for modeling with monitoring station data due to very common gaps.

Finally, Oprea et al. [48] aim to extract rules for guiding the forecasting of particulate matter ($PM_{10}$) concentration levels using Reduced Error Pruning Tree (REPTree) vs. M5P (an inductive learning algorithm). The principal component analysis is used to select the best predictors, and then 27 months of eight previous-days of $PM_{10}$, $NO_2$, $SO_2$, temperature and relative humidity data are used to predict $PM_{10}$ in Romania. The validation is performed by comparing the results of two models with the observed values, concluding that M5P provides the more accurate prediction of the short-term $PM_{10}$ concentrations ($r^2 = 0.81$, for one day ahead and $r^2 = 0.79$ for two days ahead).

### 3.2.2. Category 2: Image-Based Monitoring and Tackling Low Spatial Resolution from Non-Specific/Low Resolution Sensors

This group is composed of a set of seven papers. All of them address estimation issues only. It is quite logical that no forecasting studies are included in this category, because the principal objective of this kind of work is to increase the spatial resolution of the current pollution spreading. To do so, a first paper presents a method to improve the spatial resolution and accuracy of satellite-derived ground-level $PM_{2.5}$ by adding a geostatistical model (Random Forest-based Regression Kriging—RFRK) to the traditional geophysical model (i.e., Chemical Transport Models—CTM) [49]. This is a two-step procedure in which a Random Forest regression models the nonlinear relationship between $PM_{2.5}$ and the geographic variables (step 1) then the kriging is applied to estimate the residuals (or error) of step 1 (step 2). This work is carried out in the USA and is based on a 14 years dataset. The predictive features include CTM, satellite-derived dataset (meteorology and emissions), geographic variables (brightness of Night Time Lights—NTL, Normalized Difference Vegetation Index—NDVI, and elevation), and in situ $PM_{2.5}$. The accuracy is evaluated by comparing the RFRK to CTM-derived $PM_{2.5}$ models and using ground-based $PM_{2.5}$ monitor measurements as reference. The results show that the RFRK significantly outperforms the other models. In addition, this method has a relatively low computational cost and high flexibility to incorporate auxiliary variables. Among the three geographic variables, elevation contributes the most and NDVI contributes the less to the prediction of $PM_{2.5}$ concentrations. The main limitation of the technique is to rely on satellite images that are subject to uncertainties (e.g., data quality, completeness, and calibration).

A similar approach is proposed by Just et al. [50]. The motivation of this study is to enhance satellite-derived Aerosol Optical Depth (AOD) prediction, by applying a correction based on three different Ensemble Learning algorithms: Random Forest, Gradient Boosting, and Extreme Gradient Boosting (XGBoost) The correction is implemented by considering additional inputs as predictive factors (e.g., land use and meteorology) than AOD only. This technique is applied to predict the concentration of $PM_{2.5}$ in Northeastern USA during a period of 14 years. The accuracy of the method is assessed by comparing the coefficient of determination (predicted value/actual value) between the three models. The result shows that XGBoost outperforms slightly the other two algorithms. In addition, the study demonstrates that including land use and meteorological parameters in the algorithms improves significantly the accuracy when compared to the raw AOD. Nevertheless, a limitation of the technique is the fact that it requires lots of features (total number = 52). Furthermore, Zhan et al. [51] propose an improved version of the Gradient Boosting algorithm by considering the spatial nonlinearity between $PM_{2.5}$ and AOD and meteorology. They develop a Geographically-Weighted Gradient Boosting Machine (GW-GBM) based on spatial smoothing kernels (i.e., gaussian) to weight the optimized loss function. This model is applied to all China and uses data

of 2014. The GW-GBM provides a better coefficient of determination than the traditional GBM. The study shows that the best features in descending order are: day of year, AOD, pressure, temperature, wind direction, relative humidity, solar radiation, and precipitation.

Xu et al. [52] estimate ozone profile shapes from satellite-derived data. They develop a NN-based algorithm, called the Physics Inverse Learning Machine. The proposed method is composed of five steps. Step 1 is a clustering that gets groups of ozone profiles according to their similarity (k-means clustering procedure). Step 2 generates simulated satellite ultraviolet spectral absorption (UV spectra) of representative ozone profiles from each cluster. Step 3 consists of improving the classification effectiveness by reducing the input data through a Principal Component Analysis. In step 4 the classification model is applied for assigning an ozone profile class corresponding to a given UV spectrum. And in step 5 the ozone profile shape is scaled according to the total ozone columns. The algorithm is tested by comparing the predicted value to the observed value. 11 clusters are obtained, and the estimation error is lower than 10%. This technique can be considered an encouraging approach to predict ozone shapes based on a classification framework rather than the conventional inversion methods, even if its computational cost is relatively high.

A slightly different approach is proposed by de Hoogh et al. [53]. This work aims to build global (1 km $\times$ 1 km) and local (100 m $\times$ 100 m) models to predict $PM_{2.5}$ from AOD and $PM_{2.5}/PM_{10}$ ratio. It is based on an SVM algorithm to predict the concentration of $PM_{2.5}$ in Switzerland. The dataset is composed of 11 years of observations and a broad spectrum of features, such as: Planetary boundary layers, meteorological factors, sources of pollution, AOD, elevation, and land use. The results show that the method is able to predict $PM_{2.5}$ by using data provided by sparse monitoring stations. Another technique consists of supplementing sparse accurate station monitoring by lower fidelity dense mobile sensors, with the aim of getting a fine spatial granularity [54]. Seven Regression models are used to predict CO concentrations in Sidney, Australia. The study is divided into three stages. First, the models are built from seven years of historical data of static monitoring (15 stations) and three years of mobile monitoring. Second, the performances of each algorithm are compared. And third, field trials are conducted to validate the models. The results show that the best predictions are provided by Support Vector Regression (SVR), Decision Tree Regression (DTR), and Random Forest Regression (RFR). The validations on the field suggest that SVR has the highest spatial resolution estimation and indicates boundaries of polluted area better than the other regression models. In addition, a web application is implemented to collect data from static and mobile sensors, and to inform users.

Finally, a quite different study uses image processing to predict pollution from a video-camera analysis [55]. Again, several families of machine learning algorithms are tested and compared. The method involves four sequential steps, which are: (i) camera records, (ii) image processing, (iii) feature extraction, and (iv) machine learning classification. The objective is to assess the severity of the pollution cloud emitted by factories from 12 features characterizing the image, such as— luminosity, surface of the cloud, color, and duration of the cloud. Although all the algorithms provide a similar performance, Decision Tree is the only one to be systematically classified among the models with the best (i) robustness of the parameter setting (= easy to configure), (ii) robustness to the size of the learning set, and (iii) computational cost. This study introduces interesting performance metrics based on 'efficiency' (higher weight for the correct classification of 'critical events' than 'noncritical events') instead of 'accuracy' (same weight for any event). In addition, with the objective to tackle the issue of unbalanced classes and since the authors are more interested in critical events (black cloud), which are less represented, some 'noncritical events' are removed from the learning set in order to get a balanced representation of the classes. Nevertheless, the application of this method is limited to the assessment of the contamination produced by plants.

### 3.2.3. Category 3: Considering Land Use and Spatial Heterogeneity/Dependence

Four out of five papers classified in this category are estimation models. The most general manuscript is written by Abu Awad et al. [56]. The objective is to build a land use model to predict Black Carbon ($PM_{10}$) concentrations. The approach is divided into two steps. First, a land use regression model is improved by using a nu-Support Vector Regression (nu-SVR). Second, a generalized additive model is used to refit residuals from nu-SVR. This study is applied in New England States (USA) and exploits a 12 years dataset, which stores data from 368 monitoring stations. A broad spectrum of variables is used as features, such as: Filter analysis, spatial predictors (elevation, population density, traffic density, etc.), and temporal predictors (meteorology, planet boundary layer, etc.). The model is tested in cold and warm seasons and is assessed by comparison to actual data. Overall, the results provide a high coefficient of determination ($r^2 = 0.8$). In addition, it appears that this coefficient is significantly higher in the cold than in the warm season.

Araki et al. [57] also build a spatiotemporal model based on land use to capture interactions and non-linear relationships between pollutants and land. The study consists of comparing two algorithms—Land Use Random Forest (LURF) vs. Land Use Regression (LUR)—to estimate the concentration levels of $NO_2$. This work takes place in the region of Amagasaki, in Japan, over a period of four years. Besides land use, the authors use population, emission intensities, meteorology, satellite-derived $NO_2$, and time as features. The accuracy of LURF outperforms slightly LUR, and the coefficient of determination gets a same range of values as in the above study. LURF is a bit better than LUR, because the former allows for a non-linear modelling whereas the latter is limited to a linear relationship. Another advantage of using random forest is to produce an automatic selection of the most relevant features. The results show that the best predictors in descending order are—green area ratio, satellite-based $NO_2$, emission sources, month, highways, and meteorology. The variable population is discarded from the model. Nevertheless, the interpretation of the regression model is easier than the random forest-based model, because the former provides coefficients representing the direction and magnitude of the effects of predictor variables. The variable selection issue related to LUR is tackled in the study carried out by Beckerman et al. [58]. They develop a hybrid model that mix LUR with deletion, substitution, and addition machine learning. Thanks to this method, no more than 12 variables are used to predict the concentration of $PM_{2.5}$ and $NO_2$. The case study is applied to all California and uses data on the period 1999–2002 for $PM_{2.5}$, and 1988–2002 for $NO_2$. The overall performance of the model is fair, but the accuracy for estimating $PM_{2.5}$ is significantly lower than $NO_2$.

Brokamp et al. [31] perform a deeper analysis based on a prediction of the concertation of the chemical composition of $PM_{2.5}$. As Beckerman et al. [57], they compare the performance of a LURF versus LUR approach. The dataset is built from the measurements of 24 monitoring stations in the city of Cincinnati (USA) during the period 2001–2005. Over 50 spatial parameters are registered, which include transportation, physical features, socioeconomic characteristics, greenspace, land cover, and emission sources. Again, LURF outperforms slightly LUR. The originality of this work is to create models that predict not only the air quality, but also the individual concentration of metal components in the atmosphere from land use parameters.

Yang et al. [59] is the only study of this category that provides a forecasting model. The approach is applied in three steps. First, a clustering analysis is performed to handle the spatial heterogeneity of $PM_{2.5}$ concentrations. Several clusters are defined, based on homogeneous subareas of the city of Beijing (China). Second, input spatial features that address the spatial dependence are calculated using a Gauss vector weight function. And third, spatial dependence variables and meteorological variables are used as input features of an SVR-based model of prediction. The performance of the proposed Space-Time Support Vector Regression (STSVR) algorithm is compared to the ARIMA, traditional SVR, and NN. The best model depends on the forecasting time span. STSVR outperforms the prediction of the other models from 1 h to 12 h ahead, whereas the global SVR outperforms STSVR from 13 h to 24 h ahead. This study demonstrates that (i) the relationship between air pollutant concentrations

and other relevant variables changes over spatial areas, and (ii) as the forecasting time increases, the prediction accuracy decreases more with STSVR than SVR.

### 3.2.4. Category 4: Hybrid Models and Extreme/Deep Learning

This set of papers represents one of the rare categories in which forecasting models (10) are significantly more frequent than estimation models (2). The first estimation study consists of building a reliable model to predict $NO_2$ and $NO_x$ with a high spatiotemporal resolution and over an extended period of time [60]. This work is carried out in three steps. First, multiple non-linear (traffic, meteorology, etc.), fixed (e.g., population density) and spatial (e.g., elevation) effects are incorporated to characterize spatiotemporal variability of $NO_2$ and $NO_x$ concentration. Second, the Ensemble Learning algorithm is applied to reduce variance and uncertainty in the prediction based on step 1. Third, an optimization process is implemented by tackling possible incomplete time-varying covariates recording (traffic, meteorology, etc.) in order to get a continuous time series dataset. The prediction performance is quite high ($r^2 \approx 0.85$). Traffic density, population density, and meteorology (wind speed and temperature) account for 9–13%, 5–11%, and 7–8% of the variance, respectively. Nevertheless, the method is less accurate for the prediction of low pollution levels.

A different estimation approach is proposed by Zhang and Ding [61]. The authors use an Extreme Learning Machine (ELM) to tackle the low convergence rate and the local minimum that characterize the NN algorithms. Their ELM consists of only 2-layer NN. The first layer (hidden layer) is fixed and random (its weight does not need to be adjusted). The second layer only is trained. They estimate a broad spectrum of contaminants ($NO_2$, $NO_x$, $O_3$, $PM_{2.5}$, and $SO_2$) from meteorological and time parameters. It appears that the ELM is significantly better than NN and Multiple Linear Regression (MLR), because it provides a higher accuracy and a lower computational cost. An even more advanced study is carried by Peng et al. [62]. The authors propose an efficient non-linear machine learning algorithm for air quality forecasting (up to 48 h), which is easily updatable in 'real time' thanks to a linear solution applied to the new data. Five algorithms are compared: MLR, Multi-Layer Neural Network (MLNN), ELM, updated-MLR, and updated-ELM. This approach involves two steps and predict three types of contaminants ($O_3$, $PM_{2.5}$, and $NO_2$). Step 1 consists of the initial training stage of the algorithm. The 2nd step involves a sequential learning phase, in order to proceed with an online (daily) update of the MLR and ELM, and only seasonally (trimonthly) update of the MLNN. The results show that the updated-ELM tends to outperform the other models, in terms of $R^2$ and MAE. In addition, the linear updating applied to MLR and ELM is less costly than updating a MLNN. Nevertheless, all the models tend to underpredict extreme values (the non-linear models even more than the linear models).

Another type of forecasting involves hybrid models. This is the case of the study performed by Zhang et al. [63], who predict short and long-term CO from a hybrid model composed with Partial Least Square (PLS), for data selection, and SVM, for modelling. Overall, SVM-PLS seems more performant than traditional SVM, because both the precision and computational cost of the former is better than the latter. However, the accuracy is lower for hourly than daily forecasting. Tamas et al. [64] propose a different hybrid model to forecast air contaminants ($O_3$, $NO_2$, and $PM_{10}$), with a special focus on the critical prediction of pollution peaks. The hybrid model is composed with NN and clustering and it is compared to the Multilayer Perceptron. The proposed algorithm outperforms the traditional NN, especially in the prediction of $PM_{10}$ and $O_3$ peaks of pollution. Since performing a forecasting consists mainly of predicting a time series, Ni et al. [65] propose a hybrid model that implements a NN and an Autoregressive Integrated Moving Average (ARIMA). The concentration of $PM_{2.5}$ are predicted in Beijing (China) from meteorological parameters, chemical variables, and microblog data. Such an approach allows for a good forecast for a few hours ahead, but the bigger the time lag, the bigger the error.

A time series analysis is also used by Li and Zhu [66] to predict the AQI ($NO_2$ + $PM_{10}$ + $O_3$ + $PM_{2.5}$) 1 h ahead and from the extraction of conceptual features (randomness and fuzziness of time series data). A hybrid model based on a technique of Cloud Model Granulation (CMG) and SVR is implemented in two steps. First, CMG uses probability and fuzzy theory to extract from time series the concept features of: randomness and fuzziness of the data. And second, after extracting these features, an SVR is used for the prediction per se. The performance of CMG-SVR is slightly better than the individual application of the same techniques (CMG and SVR alone) or a NN. Nevertheless, this method is still only able to make a short-term prediction (about 1 h ahead). The fuzzy approach to forecast $PM_{10}$ hourly concentration 1 h ahead is also used by Eldakhly et al. [67]. To tackle the randomness and fuzziness of the data, the hybrid model applies, first, a chance weight value to the target variable in order to minimize an outliner point (if any) that can be used, afterward, as a support vector point during the training process. The result of this study is one of the rare examples that shows a model that outperforms ensemble learning algorithms (Boosting and Stacking).

Wang et al. [68] propose to optimize the AQI forecasting by applying a 'decomposition and ensemble' technique in three steps: (i) decomposition of the original AQI time series into a set of independent components (i.e., frequencies or IMFs), (ii) prediction of each component (by ELM), and (iii) aggregation of the forecast values of all components. The hybrid model takes every eight successive days as AQI data inputs of the ELM to forecast the ninth day. The originality of the study is to propose a decomposition of the AQI time series into two phases— by Complementary Ensemble Empirical Mode Decomposition (CEEMD), for low and medium frequencies, and by Variation Mode Decomposition (VMD), for high frequencies. This 2-phase decomposition provides a superior forecasting accuracy than 1-phase decomposition implementations. Fuzzy logic, ELM, and heuristic are put together in the model developed by Li et al. [66]. This study forecasts the AQI (based on $PM_{10}$, $PM_{2.5}$, $SO_2$, $NO_2$, CO, and $O_3$) of six cities in China. Fuzzy logic is used as a feature selection, because relevant predictive contaminants are not the same according to the city. ELM and heuristic optimization algorithms are applied for a deterministic prediction of the contaminant concentrations. This hybrid model gives a better prediction than other NN and time series (e.g., ARIMA) algorithms but it is slightly slower than the NNs.

More recently, a bunch of studies have implemented a Deep Learning approach to improve the accuracy of air pollution forecasting. For instance, Zhao et al. [69] propose a Deep Recurrent Neural Network (DRNN) method to forecast daily Air Quality Classification (AQC). This technique is divided into two steps. First, data of six pollutants are pre-processed, in order to group the concentrations into four categories or Individual Air Quality Index—IAQI (from unhealthy to good). Second, an RNN based on Long Short-Term Memory (LSTM) is used to perform the forecasting. LSTM is an RNN, which includes a memory that permits to learn the input sequence with longer time steps (e.g., problems related to time series). Despite the high expectations, the predictive accuracy of the model is not significantly better than the performance of the other tested algorithms (SVM and Ensemble Learning). In addition, the disadvantage of Deep Learning is a high computational cost and a low interpretability of the model. A more satisfactory result is obtained by Huang and Kuo [70], who also use a hybrid model based on NN and LSTM to forecast $PM_{2.5}$ concentrations 1 h ahead, by processing big data. The authors use a Convolutional Neural Network (CNN) instead of the traditional NN, because its partially connected architecture allows for a reduced training time. This optimized model seems to outperform all the principal machine learning algorithms (SVM, Random Forest, and Multilayer Perceptron). However, the relevance of such a technique still has to be confirmed over a longer time span forecast.

### 3.2.5. Category 5: Towards an Application System

With only four papers out of 46, the application category is one of the smallest. This result means that the majority of the published studies are still very theoretical. A first interesting application system is proposed by Sadiq et al. [71]. The purpose of their model is to regulate the traffic, in order to

optimize pollution level while maximize the vehicle flow. The method is based on a hybrid model that involves problem-solving, multi-agent system, and NN to estimate the concentration of $O_3$ in Marrakech-City (Morocco). The predictive features are traffic, pollutant covariates, and meteorological parameters (relative humidity, wind speed, temperature, and solar radiation). The accuracy of the model is assessed by comparing the predicted values to the measurements at the monitoring stations. The system is optimized by using the Hadoop framework that manages big data and a multi-agent software architecture. The objective is to provide the drivers with recommendations on the best path (sum of the shortest + least polluted road) after applying a search graph algorithm (Dijkstra). An app of the system is currently available.

Other studies intend to provide a public application to forecast air quality. This is the case of the work carried out by Tzima et al. [72], who develop a tool that is integratable with existing Environmental Management Systems implemented in Greece. The machine learning model implements Decision Rules to forecast pollution peaks of $O_3$ and $PM_{10}$. The obtained model is a trade-off between performance and understandability. It is less accurate than a linear regression to forecast $PM_{10}$ but it is better than the other tested models to predict $O_3$. A more advanced system is based on the operational forecasting platform Prev'Air [73]. This platform aims at forecasting maps, on a daily basis, for $O_3$, $NO_2$, and $PM_{10}$. This study implements a Ridge Regression method that reduces the errors of the forecasts (RMSE) for all the contaminants (by 35% for $O_3$, 26% for $NO_2$, and 19% for $PM_{10}$). However, the technique presents some limitations to predict pollution peaks. In that sense, it is not adapted for cities with frequent violations of air quality standards.

The last work consists of a network installation of low-cost pollution sensors [74]. The whole system is able to store data, to process these raw data for a posterior forecasting, and to present the prediction through different channels, such as mobile application and Web portal. Different machine learning methods (SVM, Regression, Model Trees, and NN) are tested, in order to identify the best algorithm to forecast the concentration levels of $O_3$, $NO_2$, and $SO_2$. It seems that the Model Trees provide the lowest RMSE. This solution is particularly relevant to tackle low density monitoring stations and could be advantageously applied in developing countries that suffer high pollution rates. Overall, the majority of the application systems implement traditional machine learning techniques.

### 3.2.6. Category 6: Nanoparticles as a New Challenge

In the last category, the studies focus on estimating the concentrations and peaks of nanoparticles. A number of years of research indicates that the levels of $PM_{2.5}$ concentrations are dropping in most of the high- and mid-income countries [75–78]. However, in the case of nanoparticles, the harm comes not from their mass (particle concentration, usually high for the larger size aerosols), which compared to micro-sized particles is minute, but the quantity (particle number, usually very high for the smaller size aerosols) entering the human blood stream and damaging the cardiovascular system. Recent study indicates that nanoparticles attack the injured blood vessels and may cause heart attacks and strokes, while long-term exposure causes vascular damage [79]. Therefore, studying these ultrafine particles (UFP) is quite important due to the increased production of very small particles by modern engines. However, not many cities have the infrastructure to monitor these complex pollutants. Therefore, Al–Dabbous et al. [80] aim to build a model that estimates the concentration of nanoparticles from accessible and routinely-monitored pollution and meteorological factors using a feed-forward Neural Network with back-propagation. First, different architectures of NN (number of layers and neurons) are tested through an empiric approach (i.e., trial and error), in order to select the models that provide the minimum error of prediction according to the features considered, then the seven best models (each one with different types and number of features) are compared, and finally, in order to assess the prediction of the models for high-pollution concentrations 75th percentile values are considered. One month of five pollutant covariates ($PM_{10}$, $SO_2$, $O_3$, $NO_x$, and CO), and/or meteorological data (wind speed, and temperature) are used in Fahaheel City (Kuwait). The model performance is assessed by comparison between the seven models (i.e., type and number

of features considered) based on the determination coefficient. The models that include meteorology have the best results $r^2 = 0.8$ (with meteorology), and $r^2 = 0.74$ (without meteorology). The accuracy of the models is slightly lower for extreme concentrations (>75th percentile values). The study shows that the concentration in nanoparticles is very sensitive to meteorological factors.

Similarly, Pandey et al. [81] aim to understand the relationships between the concentration of submicron particles and meteorological and traffic factors in order to estimate nanoparticles (particulate matter less than 1 μm—$PM_1$, and UFP) in Hangzhou (China). As in the previous study [80], a short term, three days in winter data of meteorological (temperature, relative humidity, wind speed and direction, precipitation and pressure), traffic (flow, speed) and time parameters are used. Tree-based classification models, SVM, Naïve Bayes, Bayesian Network, NN, k-NN, Ripper and RF are implemented. Correlation with PM and each predictor is performed, to know which single feature has better predicting power, and based on that the multiple parameter model is used. Data are split into two (low vs. high) and three (low vs. medium vs. high) ranges of pollution classes. Model performance is validated against the observed data (precision, area under curve—AUC) resulting in tree algorithm AUC and f-measure 1 for $PM_1$ (ADTree, RF), and for UFP 0.84 (RF). This study is a promising example in predicting nanoparticles without actual measurements. It shows that it is easier to predict $PM_1$, than UFP. Tree algorithms are able to produce almost completely accurate prediction of $PM_1$, and very good ($r^2 = 0.844$) for UFP. A binary split is better than ternary for all classifiers. Finally, weather also shows a strong relationship with $PM_1$, while traffic parameter is important for UFP.

## 3.3. Synthesis

In order to synthetize the results of this systematic review, an MCA is performed. This factorial analysis looks for the principal dimension explaining the variability between the papers and to closely examine relationships between them. The MCA dimensions are quantitative variables which summarize the qualitative variables that characterize each study. This statistical method suggests links between these qualitative variables by the observation of their proximity in the point cloud-based representation of the categories (Figure 5). Here, the total percentage of inertia is 28%, which is an acceptable value considering the fact the papers are located in a high K–J dimensional space. The *x*-axis (F1) shows an opposition between forecasting models and estimation models. On the one hand, the forecasting models seem more used to predict Air Quality Index (prediction-AQI) and are mainly based on NN (algorithm-NN) and SVM (algorithm-SVM) algorithms. They do not rely on land use predictors (land-F), but a lot on the targeted predicted parameter (target-T). Since a forecasting usually involves a time series analysis, it makes sense to use the targeted parameter as predictive feature for predicting its evolution over time. Considering the large distance between 'model-forecasting' and 'prediction-NOx', it is also possible to conclude that this kind of modelling is not used to predict the nitrogen oxides in the atmosphere. On the other hand, the estimation models constitute the main approach to predict ozone (prediction-O3) and nanoparticles (prediction-PM01) and are mostly based on Ensemble Learning algorithms (algorithm-Ensemble). Contrary to forecasting, they do not use the predicted factor as feature (target-F), because this type of model focuses on estimating the concentration of a determined pollutant from other sources of contamination or dispersion. This aspect explains the reason why atmospheric parameters (atmosphere-T), time (time-T), satellite images (image-T), and land use (land-T) tend to be located on the same side as 'model-estimation'. The *y*-axis (F2) explains less variables than the *x*-axis (F1). Nevertheless, it highlights a clear opposition between 2 peculiars algorithms that appear only in 2 different papers (algorithm-Rules and algorithm-Lazy). If we look at the data, it is possible to confirm that these two studies address distinct types of modelling (estimation vs. forecasting) and predicted parameters ($PM_{10}$ vs. AQI).

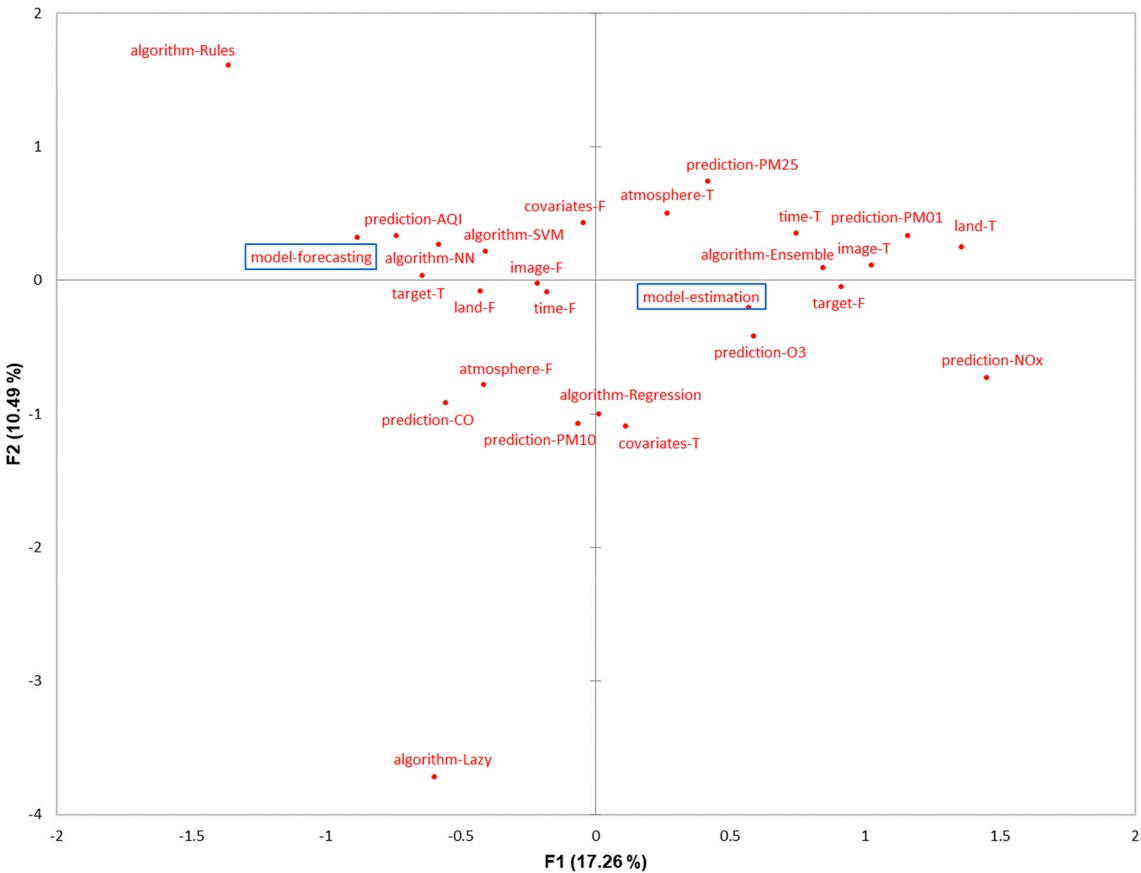

**Figure 5.** MCA representation of the point cloud of the categories that characterize the selected papers. The studies are described from nine variables (J) and 27 categories (K). Variable 1 is the type of modelling and has two categories: Estimation (model-estimation) and forecast (model-forecasting). Variable 2 is the type of algorithm and has six categories: Ensemble Learning (algorithm-Ensemble), Neural Network (algorithm-NN), Support Vector Machine (algorithm-SVM), Multiple Regression (algorithm-Regression), Decision Rules (algorithm-Rules), and Lazy methods (algorithm-Lazy). Variable 3 is the predicted parameter and has seven categories: Air Quality Index (prediction-AQI), $PM_{10}$ (prediction-PM10), $PM_{2.5}$ (prediction-PM25), nanoparticles (prediction-PM01), carbon monoxide (prediction-CO), nitrogen oxides (prediction-NOx), and ozone (prediction-O3). Variable 4 is the feature target and has two categories: Used (target-T) and not used (target-F). Variable 5 is the feature pollutant covariates and has two categories: Used (covariates-T) and not used (covariates-F). Variable 6 is the feature atmospheric parameters and has two categories: Used (atmosphere-T) and not used (atmosphere-F). Variable 7 is the feature land use and has two categories: Used (land-T) and not used (land-F). Variable 8 is the feature time and has two categories: Used (time-T) and not used (time-F). Variable 9 is the feature image processing and has two categories: Used (image-T) and not used (image-F).

Overall, more studies are dedicated to estimation (28) than forecasting (18). There are only two out of six categories in which the number of forecast motivations is higher than estimation purposes (Figure 6). This is the case of Category 4, which presents hybrid solutions to predict air contamination and includes deep and extreme learning. These techniques are usually characterized by a high performance of prediction, which is particularly adapted to tackle the uncertainty to forecast concentrations of pollutant in the near or far future. The other group is Category 5, which focuses on implementing practical applications. Intuitively, it makes more sense to develop an App or a Web portal that provides the users with a forecasting information that supports the planning of their next activities than to propose a theoretical assessment of air quality. On the contrary, the other four categories are

significantly more represented by estimation than forecasting models. Obviously, the group with the most estimations is Category 1, because it is interested in understanding the relationship between the sources of contamination and air quality. According to the MCA, it also tends to prefer models that involve a high interpretability (e.g., Regression or Decision Tree-based models) than accuracy. A similar interpretation can be made for Category 4, which includes land use parameters and other spatial heterogeneities of the environment to increase the comprehensiveness of the models. Interestingly, there are two groups (Categories 2 and 6) in which forecasting models are not represented at all. Only estimation models are built in Category 2, certainly because the priority of this kind of study is improving the spatial resolution of the current pollution spreading and not predicting it hours ahead. Finally, considering that Category 6 is a new challenge, it is natural that the only two studies on the topic focus on an estimation before carrying out a forecast. The identification of these two unexplored axes allows us to suggest where the new research should be evolving next.

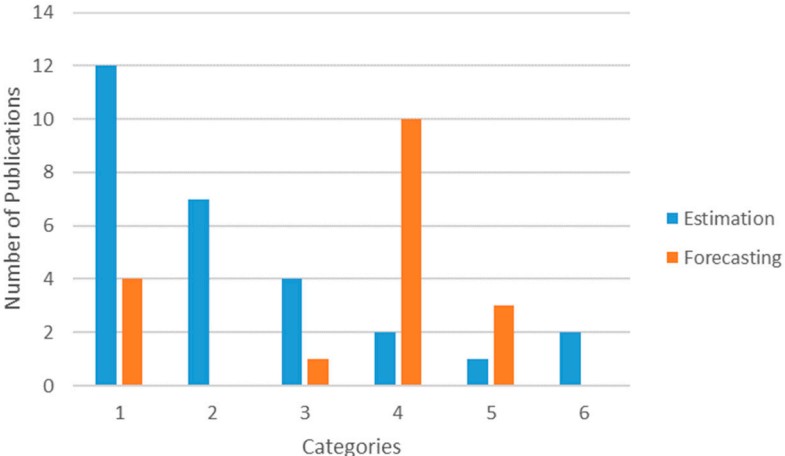

**Figure 6.** Number of types of studies (estimation vs. forecasting) per group. Category 1: Identifying relevant predictors and understanding the non-linear relationship with air pollution. Category 2: Image-based monitoring and tackling low spatial resolution from non-specific/low resolution sensors. Category 3: Considering land use and spatial heterogeneity/dependence. Category 4: Hybrid models and extreme/deep learning. Category 5: Towards an application system. Category 6: Nanoparticles as a new challenge.

Finally, the performances of all the studied models were evaluated in general per algorithm and then each algorithm performance per pollutant for estimation and forecasting studies. For the first analysis, we selected the most used algorithms: Ensemble, NN, SVM and Regression modeling to evaluate coefficients of determination resulting from all 46 studies. The best overall performance is registered for the Ensemble modeling ($r^2 = 0.79$) with a low variability in the results between different studies. Second best is the Regression modeling ($r^2 = 0.74$) with also low standard deviation. These two types of algorithms are mostly used in the estimation modeling (see Figure 5), thus resulting in a better predictive performance. SVM has also showed a good performance ($r^2 = 0.67 \pm 0.15$). This algorithm is used for the forecasting and a few studies on estimation modeling. The lowest performance was registered from NN modeling ($r^2 = 0.64 \pm 0.27$), often used in forecasting studies (see Figure 5).

In order to further identify a success of forecasting and estimation studies we separated the resulting available coefficients of determination for each pollutant. Figure 7 shows that in almost all the cases, with the exception of PM$_{2.5}$ the estimation modeling results in better accuracy than the forecasting modeling. However, the variation of the results is significantly higher in the forecasting. The estimation modeling performs quite well for CO ($r^2 = 0.76$), NO$_x$ ($r^2 = 0.78$), O$_3$ ($r^2 = 0.74$) and PM$_{2.5}$ ($r^2 = 0.72$). Meanwhile, the forecasting models reach a very high level of performance for

$PM_{2.5}$ ($r^2 = 0.74$), which is an important achievement, as CTMs often struggle with this specific pollutant [49,82]. Interestingly, the worst performance of forecasting is registered for the traffic-related pollutants CO and $NO_x$.

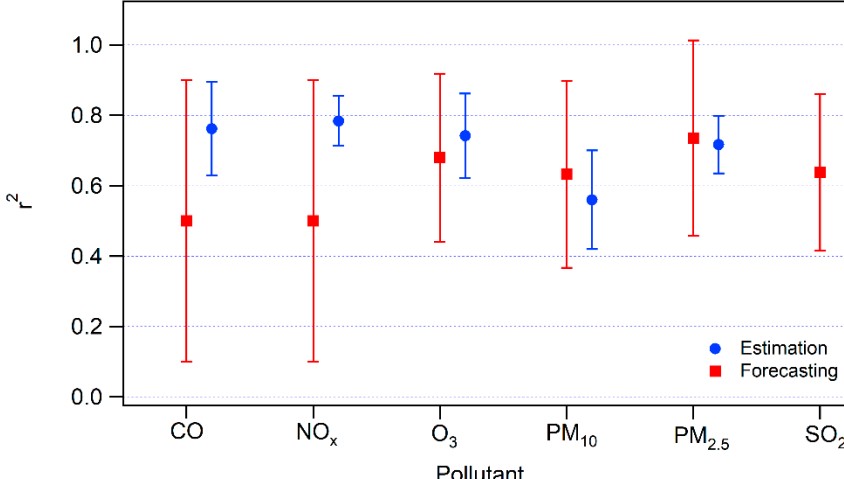

**Figure 7.** Model performance evaluated with the average and standard deviation of the available $r^2$ per pollutant of estimation and forecast studies.

## 4. Conclusions

This manuscript examines 46 selected scientific articles focusing on machine learning approach to predict air pollution. While machine learning application in atmospheric sciences has been rapidly increasing in the last two years, this growth has been restricted to Eurasian and North American continents, with only four studies in the southern hemisphere.

This systematic review shows that there are two kinds of studies that implement a machine learning approach for predicting air quality. The first class, the majority, is represented by papers that focus on an estimation of the concentration of pollutants. It mostly uses Ensemble Learning or Regression algorithms, because they provide an excellent trade-off between interpretability and performance of the model. The second class is composed of manuscripts that address forecasting problems. This kind of question is clearly treated by the use of NN and SVM techniques. The forecast prioritizes the accuracy over the interpretability, which may explain why such powerful algorithms, but also considered as a black box, are preferred. Our synthesis shows that the precision of the forecasting tends to be lower and more variable than the precision of the estimation. This fact may justify the necessity to apply methods that are computationally more demanding (e.g., deep learning) to tackle the complexity to predict the value of contaminants hours or days ahead.

Overall, it seems that machine learning is an appropriate method for the prediction of air pollution. The best example is the case of fine particulate matter. Estimating such contaminants is particularly challenging for the traditional CTM approach. With machine learning, the accuracy of both estimation and forecasting of $PM_{2.5}$ reaches one of the highest values if compared with the other pollutants. Nevertheless, the models still have reduced precision for the prediction of extreme concentrations of atmospheric pollutants. In general, the accuracy for the prediction of high pollution peaks is lower than for medium and small peaks. Also, the forecasting performance is still limited for certain chemicals, such as CO and $NO_x$. In addition, the models seem to perform better for more extreme weather conditions (e.g., fall, winter, windier, etc.). Finally, it is important to note that an increasing and dangerous form of contamination by nanoparticulate is understudied (only two papers identified in this selection). Thus, we consider that improving the models that predict pollution peaks and critical pollutants constitute the next challenges that should be addressed by future machine

learning algorithms. This conclusion should be moderated by the limitations of this systematic review, which was restricted to studies published in scientific journals only.

**Author Contributions:** Conceptualization, Y.R. and R.Z.; methodology, Y.R.; formal analysis, Y.R. and R.Z.; data curation, Y.R. and R.Z.; data analysis, Y.R. and R.Z.; writing and editing, Y.R. and R.Z.

**Funding:** This research received no external funding.

**Conflicts of Interest:** The authors declare no conflict of interest.

## Appendix A

| REFERENCE | group | model | algorithm | prediction | target | covariates | atmosphere | land | time | image |
|---|---|---|---|---|---|---|---|---|---|---|
| Lin et al., 2018 | hybrid | forecasting | SVM | AQI | T | F | F | F | F | F |
| Araki et al., 2018 | land use | estimation | Ensemble | NOx | T | F | T | T | T | T |
| Huang & Kuo, 2018 | hybrid | forecasting | NN | PM25 | T | F | T | F | F | F |
| Li & Zhu, 2018 | hybrid | forecasting | NN | AQI | T | F | F | F | F | F |
| Grange et al., 2018 | predictors | estimation | Ensemble | PM10 | F | F | T | F | T | F |
| Just et al., 2018 | spatial resolution | estimation | Ensemble | PM25 | F | F | T | T | T | T |
| Yang et al., 2018 | land use | forecasting | SVM | PM25 | T | F | T | F | F | F |
| Liu et al., 2018 | spatial resolution | estimation | Ensemble | PM25 | T | F | F | F | F | T |
| Eldakhly et al., 2018 | hybrid | forecasting | SVM | PM10 | T | F | T | F | T | F |
| Zhao et al., 2018 | hybrid | forecasting | NN | AQI | T | F | F | F | F | F |
| Zhan et al., 2018 | predictors | estimation | Ensemble | O3 | F | F | T | T | T | F |
| De Hoogh et al., 2018 | spatial resolution | estimation | SVM | PM25 | F | F | T | T | F | T |
| Chen et al., 2018 | predictors | forecasting | NN | AQI | T | F | T | F | F | F |
| Martínez-España et al., 2018 | predictors | estimation | Ensemble | O3 | F | T | F | F | F | F |
| Xu et al., 2017 | spatial resolution | estimation | NN | O3 | F | F | F | F | F | T |
| Li et al., 2017 | hybrid | estimation | Ensemble | NOx | F | F | T | T | F | F |
| Liu et al., 2017 | predictors | estimation | SVM | AQI | T | F | T | F | F | F |
| Kleine Deters et al., 2017 | predictors | estimation | Regression | PM25 | F | F | T | F | T | F |
| Hu et al., 2017 | spatial resolution | estimation | SVM | CO | T | F | F | F | T | F |
| Al-Dabbous et al., 2017 | nanoparticles | estimation | NN | PM01 | F | T | T | F | F | F |
| Peng et al., 2016 | hybrid | forecasting | NN | AQI | T | F | T | F | F | F |
| Wang et al., 2017 | hybrid | forecasting | NN | AQI | T | F | F | F | F | F |
| Zhang & Ding, 2017 | hybrid | estimation | NN | AQI | F | F | T | F | T | F |
| Ni et al., 2017 | hybrid | forecasting | NN | PM25 | T | T | T | F | F | F |
| Brokamp et al., 2017 | land use | estimation | Ensemble | PM25 | F | F | T | T | F | F |
| Abu Awad et al., 2017 | land use | estimation | SVM | PM10 | F | F | T | T | F | F |
| Zhan et al., 2017 | spatial resolution | estimation | Ensemble | PM25 | T | F | T | F | F | T |
| Sadiq et al., 2016 | application | estimation | NN | O3 | F | T | T | T | F | F |
| Carnevale et al., 2016 | predictors | estimation | Lazy | PM10 | T | T | F | F | F | F |
| Bougoudis et al., 2016 | predictors | estimation | Ensemble | AQI | T | T | F | F | F | F |
| Shaban et al., 2016 | application | forecasting | Ensemble | AQI | F | T | T | T | T | F |
| Tamas et al., 2016 | hybrid | forecasting | NN | AQI | T | F | T | F | F | F |
| Oprea et al., 2016 | predictors | forecasting | Ensemble | PM10 | T | T | T | F | F | F |
| Nieto et al., 2015 | predictors | estimation | Regression | AQI | T | F | F | F | F | F |
| Sayegh et al., 2014 | predictors | estimation | Regression | PM10 | T | T | F | F | F | F |
| Debry & Mallet, 2014 | application | forecasting | Regression | AQI | T | F | F | F | F | F |
| Singh et al., 2013 | predictors | estimation | Ensemble | AQI | T | F | T | F | F | F |
| Papaleonidas & Iliadis, 2013 | predictors | forecasting | NN | O3 | T | T | T | F | F | F |
| Beckerman et al., 2013 | land use | estimation | Regression | NOx | F | T | F | T | F | T |
| Pandey et al., 2013 | nanoparticles | estimation | Ensemble | PM01 | F | F | T | T | T | F |
| Philibert et al., 2013 | predictors | estimation | Ensemble | NOx | F | T | F | F | F | F |
| Vong et al., 2012 | predictors | forecasting | SVM | AQI | T | F | T | F | F | F |
| Yeganeh et al., 2012 | hybrid | forecasting | SVM | CO | F | T | F | F | F | F |
| Suárez Sánchez et al., 2011 | predictors | estimation | SVM | AQI | T | F | F | F | F | F |
| Gacquer et al., 2011 | spatial resolution | estimation | Ensemble | AQI | F | F | F | F | F | T |
| Tzima et al., 2011 | application | forecasting | Rules | AQI | T | F | T | F | F | F |

**Figure A1.** Source table used to perform the Multiple Correspondence Analysis (MCA).

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
