# Peer review of "Machine Learning Approaches for Outdoor Air Quality Modelling: A Systematic Review"

_applsci, doi:10.3390/app8122570_

Round 1

Reviewer 1 Report

Machine Learning Approaches for Outdoor Air Quality Modelling: a Systematic Review

This manuscript provides a literature review of machine learning approaches employed for air quality modelling. Overall, it is a great effort and in my view it is worth publishing. Below are some points which need to be addressed before the manuscript can be published.  

1. Introduction needs to be much clearer and should describe various types of modelling approaches used for air quality modelling, including dispersion modelling approaches (e.g., Gaussian, Lagrangian, Eulerian, photochemical, CFD and hybrid air quality models); and statistical modelling approaches (e.g., time series, linear regression, non-linear regression and land-use regression models). Introduction should provide a full review to educate the reader about various types of modelling approaches in details.

2. Air quality estimation and forecasting are not the only purposes of air quality modelling. Other purposes may include: assessing the impact of air pollution (e.g. human health impact, impact on vegetation, economic impact etc), evaluating the long term trend of air pollutant concentrations, understand the interaction of air pollutant with each other with weather conditions, and so on. These and other potential purposes of air quality modelling should be reviewed.

3. “Current studies show that the traditional deterministic models struggle to capture the non-linear relationship between the concentration of air pollutants and their sources of emission and dispersion. To tackle such a limitation, the most promising approach is to use statistical models based on machine learning techniques”. These two sentences are controversial – although statistical modelling approaches provide a great option for air quality modelling, it is questionable whether these approaches provide better results than process based modelling. I personally don’t agree with this statement. 

Some minor points:

4. Please carefully read the whole manuscript and check for minor grammatical mistakes and language structure. Some example are provided below.

5. The word ‘data’ is plural… in several places it is used as singular, please correct.

6. The authors say in Line 13-14: “To tackle such a limitation, the most promising approach is to use statistical models based on machine learning techniques”. In another place the author says, line 35 – 36, “Three leading approaches are used to forecast or estimate the levels of atmospheric pollution: chemical transport, statistical and machine learning”. These two statements are contradictory. In the first instance ML is a technique of statistical modelling implementation, whereas in the second instance they have been counted two different approaches. Please discuss and correct.

7. Line 31-33: “The risk populations suffering the negative effects of air pollution are children, elderly and people with respiratory and cardiovascular problems”. Please revise this sentence, the sentence means that only these people are at risk. All population is at risk however these groups are more vulnerable.

8. Line 33-34: “These effects can be avoided or diminished through raising the awareness of the air quality conditions in the cities and predicting pollution levels”. How can pollution prediction minimise the effect of pollution? Please explain or correct. Pollution monitoring or modelling tell you about air pollution levels, then you make an air quality management plan to reduce emission, as such AQ modelling doesn’t reduce air pollution levels or their impacts.   

9. Line 36-38: “On one hand, statistical models are based on a single variable linear regression, and have shown correlations between different meteorological and environmental parameters”. Regression models can be simple or multiple depending on the number of variables.

10.“Line 202-203: Category 3 is specialized on a family of studies that considers land use and/or spatial heterogeneity as feature.”. consider? Please check for such errors.

11. Line 209: “The global tendency is an increasing over the years”. Please revise this sentence and provide reference.

12. Line 211: This high value tends to maintain stable during the last two years (do you mean 3 years?).

13. Line 229: “PM10 – particulate matter under 10 μm, PM2.5 – particulate matter under 2.5 μm”, under PM10 doesn’t make sense, you need to mention ‘size’ or ‘diameter’ in the sentence.

14. 234: Two papers use more peculiar approaches related to Decision rules and Lazy methods, respectively. Delete the word ‘respectively’.

15. Line 754-755: A number of years of research indicates that the levels of PM2.5 concentrations are dropping in most of the developed countries. Provide reference.

16. Please check the reference list, especially line 952, need to give the names of all authors. e.g., “Sayegh, A.S. Comparing the Performance of Statistical Models for Predicting PM10 Concentrations. Aerosol Air Qual. Res. 2014, 10, 653–665, doi:10.4209/aaqr.2013.07.0259”

Author Response

1. Introduction needs to be much clearer and should describe various types of modelling approaches used for air quality modelling, including dispersion modelling approaches (e.g., Gaussian, Lagrangian, Eulerian, photochemical, CFD and hybrid air quality models); and statistical modelling approaches (e.g., time series, linear regression, non-linear regression and land-use regression models). Introduction should provide a full review to educate the reader about various types of modelling approaches in details.

As requested by the reviewer, the introduction was augmented and improved by including new paragraphs that present a comprehensive literature review of the main modelling approaches to predict air quality from both the chemical (lines 48-63) and statistical point of view (lines 72-90).

2. Air quality estimation and forecasting are not the only purposes of air quality modelling. Other purposes may include: assessing the impact of air pollution (e.g. human health impact, impact on vegetation, economic impact etc), evaluating the long term trend of air pollutant concentrations, understand the interaction of air pollutant with each other with weather conditions, and so on. These and other potential purposes of air quality modelling should be reviewed.

The new second paragraph (lines 37-47) added to the introduction addresses this comment. The other purposes of air quality estimation and forecasting are supported by several references.

3. “Current studies show that the traditional deterministic models struggle to capture the non-linear relationship between the concentration of air pollutants and their sources of emission and dispersion. To tackle such a limitation, the most promising approach is to use statistical models based on machine learning techniques”. These two sentences are controversial – although statistical modelling approaches provide a great option for air quality modelling, it is questionable whether these approaches provide better results than process based modelling. I personally don’t agree with this statement.

As suggested by the reviewer, we moderated a bit our statement in the first sentence of the abstract and at line 72. In particular, we specify that the traditional deterministic models tend to struggle to provide an accurate prediction of air quality in regions of complex terrain. Different references are added to support this claim [21-25 and 32-34].

4. Please carefully read the whole manuscript and check for minor grammatical mistakes and language structure. Some example are provided below.

The paper was carefully reread in order to correct the grammatical mistakes and language structure.

5. The word ‘data’ is plural… in several places it is used as singular, please correct.

This error was corrected.

6. The authors say in Line 13-14: “To tackle such a limitation, the most promising approach is to use statistical models based on machine learning techniques”. In another place the author says, line 35 – 36, “Three leading approaches are used to forecast or estimate the levels of atmospheric pollution: chemical transport, statistical and machine learning”. These two statements are contradictory. In the first instance ML is a technique of statistical modelling implementation, whereas in the second instance they have been counted two different approaches. Please discuss and correct.

You are right. Since machine learning is a statistical model, the approaches can be summarized in only two main types, which are chemical transport and machine learning. Consequently, we removed the sentence in line 35-36.

7. Line 31-33: “The risk populations suffering the negative effects of air pollution are children, elderly and people with respiratory and cardiovascular problems”. Please revise this sentence, the sentence means that only these people are at risk. All population is at risk however these groups are more vulnerable.

The sentence was corrected as follows:

‘The risk populations suffering the negative effects of air pollution the most are children, elderly and people with respiratory and cardiovascular problems.’

8. Line 33-34: “These effects can be avoided or diminished through raising the awareness of the air quality conditions in the cities and predicting pollution levels”. How can pollution prediction minimise the effect of pollution? Please explain or correct. Pollution monitoring or modelling tell you about air pollution levels, then you make an air quality management plan to reduce emission, as such AQ modelling doesn’t reduce air pollution levels or their impacts.  

We have added extra information at the end of the 1st paragraph (lines 32-36) to clarify the idea as follows:

‘These health complications can be avoided or diminished through raising the awareness of air quality conditions in urban areas, which could allow the citizens to limit their daily activities in the cases of elevated pollution episodes, by using models to forecast or estimate air quality in regions lacking monitoring data.’

9. Line 36-38: “On one hand, statistical models are based on a single variable linear regression, and have shown correlations between different meteorological and environmental parameters”. Regression models can be simple or multiple depending on the number of variables.

We agree with the reviewer. As mentioned in comment #6, this sentence was removed from the manuscript.

10. “Line 202-203: Category 3 is specialized on a family of studies that considers land use and/or spatial heterogeneity as feature.”.. consider? Please check for such errors.

We believe the sentence is correct, because the subject is ‘family’ not ‘studies’.

11. Line 209: “The global tendency is an increasing over the years”. Please revise this sentence and provide reference.

There is no reference to add, because we are simply describing the profile of the curve presented in Figure 2, in which we can note an increase in the number of publications over the years.

However, the sentence was slightly changed to avoid confusion as follows:

‘The number of studies in this field shows an increasing trend over the last eight years.’

12. Line 211: This high value tends to maintain stable during the last two years (do you mean 3 years?).

No, the graph plotted in Figure 2 shows a stabilization of the curve over the last two years (2017 and 2018).

13. Line 229: “PM10 – particulate matter under 10 μm,  PM2.5 – particulate matter under 2.5 μm”, under PM10 doesn’t make sense, you need to mention ‘size’ or ‘diameter’ in the sentence.

The error indicated in the caption of figure 3 (lines 269-270) was corrected as follows:

‘The pollutant studies were considered per continent: O3 – ozone, PM10 – particulate matter with aerodynamic diameters  ≤ 10 µm, BC – black carbon, PM2.5 – particulate matter with aerodynamic diameters ≤ 2.5 µm, AQI – Air Quality Index, NO2 – nitrogen dioxide, NOx – nitrogen oxides (NO+NO2), SO2 – sulfur dioxide, CO – carbon dioxide, NanoPM – nano particles (particles with diameter in nm).’

14. 234: Two papers use more peculiar approaches related to Decision rules and Lazy methods, respectively. Delete the word ‘respectively’.

The word ‘respectively’ was removed.

15. Line 754-755: A number of years of research indicates that the levels of PM2.5 concentrations are dropping in most of the developed countries. Provide reference.

4 references [75-78] have been added at line 794.

16. Please check the reference list, especially line 952, need to give the names of all authors. e.g., “Sayegh, A.S. Comparing the Performance of Statistical Models for Predicting PM10 Concentrations. Aerosol Air Qual. Res. 2014, 10, 653–665, doi:10.4209/aaqr.2013.07.0259”.

The error was corrected.

Reviewer 2 Report

Review of “Machine Learning Approaches for Outdoor Air Quality Modelling: a Systematic Review”

General comments:

This paper presents a review of application of machine learning approaches for air quality modelling. Not only descriptive analysis of 46 journal papers by categories are shown, but also MAC is applied to analyze these papers. This paper is well organized, and I shall recommend its publication after minor revision.

Specific comments:

1. It is suggested adding data table mentioned in line 148 and complete disjunctive table to supplementary materials.

2. I suggest adding a brief description of PRISMA guideline.

3. Line 61: is conform -> conforms

4. Line 331-332: Rewrite “In each group …… on forecasting models”

5. Line 360-362: Rewrite “While the performance …… mountain sites”

6. Line 800-802: Rewrite “On one hand …… algorithms”

Author Response

1. It is suggested adding data table mentioned in line 148 and complete disjunctive table to supplementary materials.

The table mentioned in line 148 is added as supplementary material (see Appendix A), because it is created by the authors. Nevertheless, the disjunctive table cannot be provided, because it is automatically produced by the program (R), from the aforementioned table, and hidden from the users.

2. I suggest adding a brief description of PRISMA guideline.

A brief description of the PRISMA guideline and its application to our systematic review is added in lines 123-126. It is also mentioned that the flow diagram of the search method depicted in Figure 1 is based on the PRISMA recommendations.

3. Line 61: is conform -> conforms

This error was corrected.

4. Line 331-332: Rewrite “In each group …… on forecasting models”

This sentence was rephrased in order to improve its grammatical structure.

5. Line 360-362: Rewrite “While the performance …… mountain sites”

This sentence was also rephrased in order to improve its grammatical structure.

6. Line 800-802: Rewrite “On one hand …… algorithms”

We do not understand why you ask to rewrite this sentence. If it is because of the ‘On one hand’, we wrote ‘On the other hand’ at line 843.
